# The role of *Phragmites* on the $CH_4$ and $CO_2$ fluxes in a minerotrophic peatland in Southwest Germany

Merit van den Berg[1], Joachim Ingwersen[1], Marc Lamers[1], and Thilo Streck[1]

[1]Institute of Soil Science and Land Evaluation, University of Hohenheim, Stuttgart, Germany

5     *Correspondence to:* Merit van den Berg (merit.vandenberg@uni-hohenheim.de)

**Abstract.** Peatlands are interesting as carbon storage option, but are also natural emitters of the greenhouse gas methane ($CH_4$). *Phragmites* peatlands are particularly interesting due to the global abundancy of this wetland plant (*Phragmites australis* (Cav.) Trin. ex Steud.) and the highly efficient internal gas transport mechanism, which is called Humidity Induced Convection (HIC). The research aim was to (1) clarify how this plant-mediated gas transport influences the $CH_4$ fluxes, (2) which other environmental variables influence the $CO_2$ and $CH_4$ fluxes, and (3) whether *Phragmites* peatlands are a net source or sink of greenhouse gases. $CO_2$ and $CH_4$ fluxes were measured with the eddy covariance technique within a *Phragmites*-dominated fen in Southwest Germany. One year of flux data (March 2013 to February 2014) shows very clear diurnal and seasonal patterns for both $CO_2$ and $CH_4$. The diurnal pattern of $CH_4$ fluxes was only visible when living green reed was present. In August the diurnal cycle of $CH_4$ was most distinct, with 11-times higher midday fluxes (15.7 mg $CH_4$ m$^{-2}$ h$^{-1}$) than night fluxes (1.41 mg $CH_4$ m$^{-2}$ h$^{-1}$). This diurnal cycle correlates the highest with global radiation, which suggest a high influence of the plants on the $CH_4$ flux. But if the cause would be the HIC, it is expected that relative humidity would correlate stronger with $CH_4$ flux. Therefore, we conclude that in addition to HIC, at least one additional mechanism must be involved in the creation of the convective flow within the *Phragmites* plants. Overall, the fen was a sink for carbon and greenhouse gases in the measured year, with a total carbon uptake of 221 g C m$^{-2}$ yr$^{-1}$ (26% of the total assimilated carbon). The net uptake of greenhouse gases was 52 g $CO_2$-eq m$^{-2}$ yr$^{-1}$, which is summed from an uptake of $CO_2$ of 894 g $CO_2$-eq m$^{-2}$ yr$^{-1}$ and a release of $CH_4$ of 842 g $CO_2$-eq m$^{-2}$ yr$^{-1}$.

**Key words:** Greenhouse gases, fen, common reed, plant-mediated gas transport, eddy covariance

## 1. Introduction

Approximately one third of the world's soil carbon is stored in peatlands, although they cover only 3% of earth's total land surface (Lai, 2009). Therefore, peatland conservation or restoration as a climate change mitigation option has recently gained much attention (Bonn *et al*., 2014). Apart from the positive effect of carbon storage, peatlands are also natural emitters of methane. Methane is a 28-times stronger greenhouse gas than carbon dioxide calculated over a 100-year cycle (IPCC, 2013). Estimates of methane emissions from peatlands range between 30-50 Tg yr$^{-1}$ worldwide (Roulet, 2000). There is a high variation in methane emissions. This variability, however, and all underlying processes are not yet well understood (Hendriks *et al*., 2010; Segers, 1998). It is therefore essential to gain more knowledge about the role of methane in the greenhouse gas budgets of peatlands.

In wetland ecosystems, methane can be transported from the soil to the atmosphere via diffusion, ebullition and via aerenchyma of roots and stems of vascular plants (Moore, 1994; Le Mer and Roger, 2001; Hendriks *et al*., 2010). The largest part of the methane produced in peatlands is directly oxidized in the soil (Le Mer and Roger, 2001; Brix *et al.*, 2001; Lai, 2009). The extent of oxidation depends on the gas transport pathway and is highly dependent on the position of the water table (Moore, 1994; Le Mer and Roger, 2001; Brix *et al*., 2001; Lai, 2009) and the presence of vascular wetland plants (Grünfeld and Brix, 1999; Hendriks *et al*., 2010). Compared to other wetland plants, *Phragmites australis* (common reed) appears to have a high ability to transport gases between the soil and atmosphere (Salhani and Stengel, 2001).

The gas exchange within *Phragmites* plants takes place via convective flow through the culm. Currently it is believed that this transport originates from creating a humidity-induced pressure gradient between the internal culm and atmosphere (Armstrong and Armstrong, 1990; Armstrong and Armstrong, 1991; Armstrong *et al*., 1996b; Afreen *et al*., 2007). The pores (stomata) in the leaf sheaths of *Phragmites* are more resistant to pressure flow than against gas diffusion. Due to the higher humidity in the internal culm of the reed, $O_2$ and $N_2$ concentrations inside the plant are diluted. Therefore $O_2$ and $N_2$ are transported along the concentration gradient from the atmosphere into the sheaths and a higher pressure is created. This causes an airflow from the green living reed stems to the rhizomes and goes back to the atmosphere via dead/broken stems that are still connected to the rhizomes. This mechanism is more than 5 times as efficient as diffusion (Brix *et al*., 2001) and is also found in other wetland plants (e.g. *Nuphar*, *Eleocharis*, *Nelumbo* and *Typha*) that have a submerged rhizome system (Dacey and Klug, 1979; Dacey, 1987; Brix *et al*., 1992; Bendix *et al*., 1994). In a *Phragmites* dominated wetland, 70% of the produced methane is transported through the plants (Brix, 1989). This means that methane emissions should be highly dependent on this transport mechanism. Apart from this potential influence of HIC on the methane fluxes, *Phragmites* wetlands can also accrete large amounts of carbon in the soil due to the high annual primary production compared to other wetland plants (Brix *et al*., 2001; Zhou *et al*., 2009).

Several studies on methane emissions (Kim *et al*., 1998a; van der Nat and Middelburg, 2000) and $CO_2$ emissions (Zhou *et al*., 2009) from *Phragmites*-dominated wetlands have been published. Most of them used the closed chamber method. Despite *Phragmites australis* being the most abundant wetland species on earth, to date, the eddy covariance (EC) technique

has only been used at two study sites: Kim *et al.* (1998a) performed $CH_4$ flux measurements in a fen in Nebraska, USA, and Zhou *et al.* (2009) measured $CO_2$ fluxes from a *Phragmites* wetland in Northeast China. To our knowledge, there exist no EC $CO_2$ and $CH_4$ flux data from European *Phragmites* wetlands.

To contribute to a better understanding of the role of *Phragmites* on $CH_4$ and $CO_2$ fluxes, flux measurements were done in the minerotrophic peatland "Federseemoor" located in Southwest Germany. With the eddy covariance method, we were able to measure the net ecosystem exchange of $CH_4$ and $CO_2$ in high temporal resolution. This made it possible to detect the influence of the plant-mediated gas transport of *Phragmites* on the $CH_4$ fluxes and to evaluate the role *Phragmites* peatland plays in climate change. We recorded the diurnal and seasonal patterns of these fluxes, evaluated the impact of environmental variables on the fluxes, and determined the carbon and greenhouse gas budgets of this ecosystem. In this paper, we present the results from a measurement period of one year, from March 2013 to February 2014.

## 2.    Materials & Methods

### 2.1 Study site

The study was conducted in the Federseemoor (48.092°N, 9.636°E), a peatland with an area of 30 $km^2$, that is located in the region Upper Swabia in Southwest Germany. This region is characterized by its moraines and is located on the edge of a high rainfall zone caused by the Alps. Therefore, with a yearly precipitation around 800 mm and an average temperature of 7.1 °C, the area is wetter and colder than the average for Germany. The Federseemoor has developed via natural terrestrialisation from a proglacial lake of 30 $km^2$ that was formed after the last ice age. The lake diminished to a size of 12 $km^2$, surrounded by fen and bog. Between the years of 1787 and 1808, the lake size was further reduced by drainage activities to a size of 1.4 $km^2$. The resulting 11 $km^2$ of reclaimed land was meant for agricultural purposes, but appeared to be unprofitable. Natural vegetation started to develop and today it is a nature conservation area, mainly consisting of fen but also containing transitional bog and wooded swamp.

The lake Federsee is completely surrounded by *Phragmites* vegetation, with a total area of 2.2 $km^2$ and a density of approximately 70 living shoots per $m^2$. To the northeast of the lake, in the middle of the reed, an eddy covariance (EC) tower was constructed (Fig. 1).

### 2.2 Field measurements

The location of the EC tower was selected so that only reed vegetation is within 200 m distance of the tower (the potential footprint). An LI-7700 open-path $CH_4$ gas analyser (LI-COR Inc., USA), an LI-7200 enclosed-path $CO_2/H_2O$ gas analyser (LI-COR Inc., USA) and a WindMaster Pro sonic anemometer (GILL Instruments Limited Inc., UK) were installed at a height of 6 m, twice as high as the reed canopy. Molar mixing ratio/mass density of the gases and wind speed in three directions were measured at a frequency of 10 Hz. The LI-7700 is able to detect concentrations to 50 ppm $CH_4$, and was

calibrated for the concentration range 0-40 ppm $CH_4$ by the manufacturer in June 2012. The LI-7200 was calibrated up to 740 ppm by the manufacturer in July 2012. The random error computed by the EddyPro software (see below) of the $CH_4$ flux was around 15% and for $CO_2$ flux 20%.

Air temperature and air relative humidity (HMP155, Vaisala Inc., Finland) and incoming and outgoing shortwave and longwave radiation (CNR4, Kipp & Zonen Inc., The Netherlands) were measured at a height of 6m. Soil temperature was measured in 5, 15 and 30cm depth (LI-COR Inc., USA). Groundwater table was continuously measured with a groundwater datalogger (MiniDiver, Eijkelkamp Agrisearch Equipment Inc., The Netherlands). Rainfall (TR-525USW, Texas Instruments Inc., USA) was measured above the canopy (at a height of 3m). These environmental variables were measured every minute with exception of the water table height, which was measured every 30 minutes. Vegetation height was measured weekly.

**2.3 Flux computation**

Fluxes from 01-03-2013 until 28-02-2014 were calculated with an averaging interval of 30 minutes using the software EddyPro version 5.1 (LI-COR Inc., USA). With this software, corrections are applied to average wind directions and gas concentrations and fluxes.

The declination of the angle-of-attack, caused by the shape of the anemometer, was corrected according to Nakai and Shimoyama (2012). To correct the tilt of the anemometer or angle of the mean horizontal wind, the double rotation method was applied (Wilczak *et al.*, 2001). To convert from $CH_4$ mass density to molar concentrations, data were compensated for density fluctuations due to changes in water vapour and temperature (Webb *et al.*, 1980). This does not apply to $CO_2$/$H_2O$ gases, since the temperature and pressure are maintained constant in the enclosed path gas analyser. Therefore mixing ratios were used for the flux calculation.

The calculated fluxes were checked for quality by means of the 1-9 flagging system of Foken *et al.* (2004). Only fluxes with quality flags 1-6 were used for further data processing. Outliers were filtered out by removing fluxes that were more than 4 times the median within a time window of 6 hours and with 6 or more data points within this time window. Because of the often low turbulent conditions and stable stratification during the night, night fluxes with an average friction velocity <0.15 m/s were not considered in the data analysis.

After analysing the footprint, it appeared that all fluxes are within 200 m distance from the tower. The distance of the flux is an output of EddyPro by the method of Kljun *et al.* (2004), with the criteria that 90% of the measured gas concentration has its source within that distance. This means that at our site only reed vegetation is within the measured footprint.

**2.4 Gap filling**

Due to technical failures and discarding data due to flux quality criteria, 46% of the $CH_4$ data and 35% of the $CO_2$ data were missing. Gaps were filled with the online tool provided by the Max Planck Institute for Biogeochemistry in Jena (Germany) (http://www.bgc-jena.mpg.de/~MDIwork/eddyproc/). This tool uses the look-up table method described by Falge *et al.*

(2001) and Reichstein *et al.* (2005). This method was developed to fill $CO_2$ flux gaps. It uses the correlation of $CO_2$ fluxes with meteorological variables like global radiation, ambient temperature and vapour pressure deficit.

To date there is no established gap filling method for $CH_4$. Nevertheless, we found clear correlations between $CH_4$ fluxes and global radiation, temperature and relative humidity in our data. Therefore, we used the same gap filling method for $CH_4$ as for $CO_2$.

In the case of power or data logger failure, meteorological data were taken from a meteorological station run by the Federal State of Baden-Württemberg (LUBW) at a 2.2 km distance from the EC station.

Even with these data, the online tool still lacked sufficient data to properly fill a 2 month data gap that was caused by insufficient solar power within the time period 24-11-2013 to 30-01-2014. This was due to the maximum time window (14 days) that the tool uses. Therefore, a look-up table was made manually, to fill this data gap. Global radiation classes with an interval of 100 W m$^{-2}$ (from 0 to 800 W m$^{-2}$) and ambient temperature classes with an interval of 4 °C (from -10 to 18 °C) were created. Per combined class of temperature and global radiation the average flux was used from the available data from November and February with the same class. Gaps in the look-up table were filled by linear interpolation. To estimate how reliable these gap filled data are, an artificial gap was created for the month May and was filled based on a look-up table created with data from April. The difference between observed and estimated data was on average +30%. Therefore, data from the filled gap of December to January are only used for the annual carbon budget estimation, but not for statistical analyses.

**2.5 Separating NEE**

The $CO_2$ fluxes measured with eddy covariance are the net ecosystem exchange (NEE). By definition, this is the gross ecosystem production (GEP) minus the ecosystem respiration ($R_{eco}$). Separating $R_{eco}$ and GEP from NEE is done by using a simplified form of the regression model of Lloyd and Taylor (1994) (Eq. (1)) which describes the relation between respiration and temperature. The $Q_{10}$ value was derived by regressing the average $CO_2$ night flux ($R_{eco}$) on the average night soil temperature (T) (in °C) for every 2 months (more or less the duration of the different plant development stages) over the vegetation period (March – October):

$$R_{eco} = R_0 * Q_{10}^{T/10} \tag{1}$$

$R_0$ is the respiration when *T* is equal to zero. Only average night fluxes with 6 or more data points were considered and night is determined as the period where global radiation < 10 W m$^{-2}$. The $Q_{10}$ values of the ecosystem respiration ($R_{eco}$) obtained (between 1.53 and 3.93, see Fig. 2) were used to calculate the $R_{eco}$ for every half an hour during the day within the vegetation period. This was done by using the average $CO_2$ night flux as the reference respiration ($R_{ref}$) and the corresponding average night temperature as reference temperature ($T_{ref}$). The difference between $T_{ref}$ and the daytime temperature was used to calculate the difference between $R_{ref}$ in daytime $R_{eco}$. The GEP was then calculated by subtracting $R_{eco}$ from NEE for every

half an hour during the day. Only for the period March-April we found no dependency between $CO_2$ night fluxes and soil temperature. This is very likely due to the low soil temperature during this period (mostly below 4 °C). For these 2 months, daytime respiration was estimated by taking the average night $CO_2$ flux from that same day.

## 2.6 Statistics

Biserial (Pearson's) and partial correlation coefficients were calculated to explore the relationship between measured gas fluxes and environmental factors. For this we used the data at half-hourly resolution. Because the samples are autocorrelated in time and hence not independent, no confidence intervals were inferred and correlation coefficients will be interpreted solely in a descriptive manner.

The impact of environmental factors on the gas fluxes was analysed by polynomial regression models, making it possible to model also nonlinear relationships between environmental factors and gas fluxes. Because of the autocorrelative structure in the data series regression was applied in the framework of the ARIMA (auto-regressive integrated moving average) Box-Jenkins modeling approach. To achieve stationarity (constant expectation and variance) all data (gas fluxes and potential regressor variables) were differenced prior to the analysis. For the ARIMA analysis we used the daily averaged data measured in the vegetation period from 14 May to 31 October.

All statistical analyses were performed with the software PASW Statistics version 18.0 (SPSS Inc, released in 2009).

## 3. Results

### 3.1 Seasonal pattern in gas fluxes and environmental variables

The daily averages of $CO_2$ and $CH_4$ fluxes are presented in Fig. 3, together with the most important environmental variables. Only data to the 25-11-2013 are shown, because of the high amount of missing data after this date. The northern hemisphere´s seasonal pattern is clearly visible in temperature and global radiation. These variables show the highest values in July, with average air and soil temperatures of 19 °C and 14 °C, respectively, and daily averaged global radiation of 278 W m$^{-2}$. During the whole year, the water table never dropped below the soil surface, which means that the soil was water-saturated all the time.

The increase of temperature and global radiation at the beginning of the season initiated reed growth, starting by the 30$^{th}$ of April. From May, the reed plants assimilated $CO_2$ and daily $CO_2$ fluxes became clearly negative. At the same time $CH_4$ fluxes rapidly increased. With green reed present, both $CO_2$ and $CH_4$ daily fluxes mainly follow global radiation (see Sect. 3.3), but in an inverse manner. This suggests a high influence of the vegetation on both fluxes. The highest $CO_2$ fluxes were measured in July, the month with the highest temperatures and maximum reed height (260 cm). In July, the average flux was -17.5 g $CO_2$ m$^{-2}$ d$^{-1}$. The highest fluxes of $CH_4$ were measured in August, with an average of 0.151 g $CH_4$ m$^{-2}$ d$^{-1}$. From early October, when the reed entered the senescence stage, fluxes became smaller ($CO_2$ positive) and on average there was no

longer uptake of $CO_2$. The lowest fluxes were measured in winter (November to February, data not shown), with an average release of 2.72 g $CO_2$ m$^{-2}$ d$^{-1}$ and 0.044 g $CH_4$ m$^{-2}$ d$^{-1}$.

## 3.2 Diurnal pattern

To see how diurnal cycles of both $CO_2$ and $CH_4$ fluxes change over the season, the monthly averaged diurnal fluxes of both gases are presented in hourly resolution (Fig. 4). $CO_2$ shows a weak diurnal pattern in the months of March and April, while there is no clear pattern visible for $CH_4$. From May on, when new reed was present, a distinct diurnal pattern was established for both gases, with the highest negative fluxes for $CO_2$ and highest positive fluxes for $CH_4$ between 10:00h and 13:00h. Over the whole growing season, the daily maximum for $CH_4$ and $CO_2$ flux was on average 15 and 30 minutes, respectively, earlier than the radiation maximum. The highest midday to night difference for $CH_4$ was observed in August with, on average, a midday flux of 15.7 mg $CH_4$ m$^{-2}$ h$^{-1}$ and a night flux of 1.41 mg $CH_4$ m$^{-2}$ h$^{-1}$. These values differ by a factor of 11. The highest uptake of $CO_2$ around noon occurred in July (2.36 g $CO_2$ m$^{-2}$ h$^{-1}$). Also in this month the highest night flux was observed, on average a release of 0.629 g $CO_2$ m$^{-2}$ h$^{-1}$.

The diurnal pattern of $CO_2$ disappeared in October. From November on, only positive fluxes were measured. The diurnal pattern of $CH_4$ continued one month longer and almost vanished in November.

## 3.3 Factors affecting the fluxes during growing season

Fig. 5 shows the results of the partial correlation analysis of the half hourly data for the growing period (May-October). $CH_4$ flux shows the highest correlation with global radiation, followed by relative humidity and air temperature. The biserial correlation between $CH_4$ fluxes and global radiation changes very little, no matter which other factor is partialled out. This suggests that global radiation is the most important factor influencing $CH_4$ fluxes. The high biserial correlation of $CH_4$ flux with relative humidity and air temperature, decreases considerably when global radiation is partialled out. This means that the correlations of relative humidity and air temperature with the $CH_4$ flux are based to a large extent on their correlation with global radiation. It is notable that the correlation of the $CH_4$ flux with soil temperature is small. The correlation even becomes negative when air temperature is partialled out. During the winter period, results differ: $CH_4$ flux correlates most with soil temperature (r=0.371), followed by water table height (r=0.222) (data not shown).

The correlation table for $CO_2$ fluxes shows the same pattern, but inverse of $CH_4$, except that correlations with air and soil temperature are higher than those of $CH_4$.

The impact of environmental factors on the daily fluxes of $CH_4$ and $CO_2$ fluxes was evaluated by regression analysis in the framework of the ARIMA approach. An ARIMA(0,1,1) model was found to be suited to model the flux time series of both $CH_4$ and $CO_2$ (Table 1). Global radiation turned out to be the only regressor with a statistically significant impact (P < 0.05) on the $CH_4$ fluxes, and global radiation and soil temperature on the $CO_2$ fluxes. A second order polynomial model describes the relation between global radiation and $CH_4$ flux as well as $CO_2$ flux best. In the $CO_2$ model, the addition of soil

temperature as a linear regression term was giving the best model results. Other environmental factors, such as relative humidity and air temperature, also covary with the fluxes, but their possible impact on the fluxes cannot be determined, because it is screened due to their correlation with global radiation and soil temperature. After differencing the data, the resulting models for $CH_4$ and $CO_2$, are given by Equations (2) and (3)

$$\Delta CH_4 flux_t = \beta_1 * \Delta Rg_t + \beta_2 * \Delta Rg_t^2 + e_t - \theta e_{t-1} \tag{2}$$

$$\Delta CO_2 flux_t = \beta_1 * \Delta Tsoil_t + \beta_2 * \Delta Rg_t + \beta_3 * \Delta Rg_t^2 + e_t - \theta * e_{t-1} \tag{3}$$

where $\Delta$ is the differencing operator ((e.g., $\Delta CH_{4,t} = CH_{4,t} - CH_{4,t-1}$), $\beta$ a regression coefficient, $\theta$ the weight of the moving average (MA) term and $e_t$ the residual error term that is assumed to be independently normally distributed (white noise). The coefficient of determination ($R^2$) of the $CH_4$ model with differenced data is 0.79. Without the error term ($e_t - \theta e_{t-1}$), $R^2$ is 0.76. In case of $CO_2$, the model performance is much lower ($R^2 = 0.47$ with error term and $R^2 = 0.26$ without error term). For comparison, when only the regression part of the model is run with none-differenced data ($CH_4 flux_t = constant + \beta * Rg_t$) the coefficient of determination became 0.57. In case of $CO_2$ the respective value is 0.61. Fig. 6 gives an impression of the model performance over time, again without making use of the error terms for the predictions.

The high correlation of global radiation with the $CH_4$ fluxes suggests an influence of the plants on the fluxes. The internal gas transport mechanism of *Phragmites* (humidity-induced convection) is expected to be regulated by the stomata (influenced by radiation) and humidity differences between the atmosphere and plant culm. Therefore, we selected small intervals of global radiation within a temperature range between 10–20 °C and evaluated the correlation between relative humidity and $CH_4$ fluxes within these intervals (see Fig. 7). Only with low radiation (3-10 W m$^{-2}$) there is a clear negative correlation. With higher radiation intensities (293-300 and 593-600 W m$^{-2}$) the correlation disappears.

**3.4 Carbon and greenhouse gas balance**

Fig. 8A shows the monthly cumulative fluxes of $R_{eco}$, GEP and $CH_4$. The highest fluxes for $R_{eco}$ and GEP were measured in July and for $CH_4$ in August (3.5 g C m$^{-2}$). The contribution of $CH_4$, however, to the overall carbon flux is minor. From June to September, the contribution of GEP was higher than that of $R_{eco}$ plus $CH_4$, resulting in a net carbon uptake during these months. This uptake more than compensates the net release of carbon in the other months. The net yearly $CO_2$ uptake was 894 g m$^{-2}$ a$^{-1}$ and the $CH_4$ emission was 30 g m$^{-2}$ a$^{-1}$ (see Table 2). This leads to a net annual uptake of carbon of 221 g C m$^{-2}$ by the reed ecosystem, corresponding to 26% of the GEP. $CH_4$ plays a minor role in the carbon balance, but having a global warming potential of 28 (GWP$_{100}$, IPCC 2013), it heavily affects the greenhouse gas balance (see Fig. 8B). With an uptake of 52 g $CO_2$-eq m$^{-2}$ yr$^{-1}$, the ecosystem is a minor greenhouse gas sink (see Table 2).

4. **Discussion**

## 4.1 CH$_4$ fluxes and plant mediated gas transport

In the period that the above-ground plant parts were alive and green, we observed a distinct diurnal pattern in the CH$_4$ fluxes. The highest emission was observed around noon and the lowest during the night. Similar diurnal CH$_4$ flux patterns from *Phragmites*-dominated wetlands were reported by Kim *et al.* (1998b) who used eddy covariance (EC), and by van der Nat *et al.* (1998) and Grünfeld and Brix (1999) who performed studies with closed chambers. The observed pattern can be explained by the gas transport mechanism within the culm of the *Phragmites* plants. This mechanism is described by Armstrong and Armstrong (1990, 1991) and Armstrong *et al*. (1992, 1996a, 1996b) as humidity-induced convection (HIC). According to these publications, a convective flow is generated due to an elevated air pressure in the plant stem caused by a humidity gradient (regulated by the stomata) between the inner part of the leaf sheaths and the atmosphere. The higher pressure creates an air flow through the entire stem and rhizomes which is vented via old (broken) stems. This process starts after sunrise, is at its optimum in the early afternoon, and decreases until sunset (Brix *et al*., 2001). During the night, when stomata are closed, gas transport in the stems solely takes place via diffusion. Arkebauer *et al.* (2001) measured air pressure in stems of *Phragmites* in the field, and observed the same diurnal pattern as we found in the CH$_4$ flux data. Brix *et al*. (1992) found the same pattern with four different wetland plants (incl. *Phragmites*). They both showed that stem pressure (and convective flow in Brix *et al*.) correlates with radiation, air temperature and relative humidity. These correlations with HIC were also found in lab experiments (Armstrong and Armstrong, 1991). We found only a strong correlation of CH$_4$ fluxes with global radiation during the growing season. The correlations we found with air temperature and relative humidity can also be explained by the dependency of these variables on global radiation. It is unexpected that the correlation with relative humidity is not prominent, since this is the driving factor behind HIC. Armstrong and Armstrong (1991) found that convective flow and relative humidity were negatively correlated with a convective flow close to zero with a relative humidity of 100%. This lab experiment, however, was carried out with a very low, constant light intensity (4.4 W m$^{-2}$). Sunlight intensity can be more than 200 times higher. When we selected our measured data within the same light intensity range (Rg 3-10 W m$^{-2}$), we found exactly the same negative correlation between CH$_4$ fluxes and relative humidity as Armstrong and Armstrong (1991). With higher light intensities, however, the correlation vanished. In that same study and in another study (Armstrong and Armstrong, 1990) a correlation was found between photosynthetically active radiation (PAR) and air flow within the plant stem. Radiation can create a temperature difference between the stem and air, this increases the pressure inside the stem compared to the air pressure, which can create a convective flow as well. This phenomenon is called thermal transpiration, but in *Phragmites* the contribution is believed to be small (Armstrong and Armstrong, 1991; Armstrong *et al*., 1996a). It also appears that convective flow increases much more with PAR than with infrared radiation (Armstrong and Armstrong, 1990), which speaks against the thermal transpiration hypothesis. The strong correlation between global radiation and CH$_4$ flux that we observed and the fact that the dominant role of radiation was confirmed in the ARIMA analysis suggests that a mechanism related to stomatal control or photosynthesis might play a role in the creation of a convective flow. But the question is still how. Based on our data we cannot give an answer to this question.

We found the highest midday-night difference in the month of August when the reed was fully grown. On average, midday emissions during this month were 11 times higher than at night. This is more than 2 times higher than the highest difference Kim *et al.* (1998b) found in a *Phragmites*-dominated marsh in Nebraska. In a lab experiment by Grünfeld and Brix (1999), midday and night fluxes differed by a factor 2.5, which is also much lower than in our study. The reason for this deviation might be the density of the *Phragmites* plants for which convective flow is expected to be directly proportional. At our site, the density of living green *Phragmites* plants is almost twice as high (68 m$^{-2}$) as in the prairie marsh in Nebraska (Kim *et al.*, 1998b).

The question remains whether the overall CH$_4$ flux increases or decreases due to the presence of living green reed. In our data, we found a very clear increase in the daily CH$_4$ flux after the beginning of reed growth. Soil temperature also increased in the month May, but not in proportion to the CH$_4$ flux. An increase of the CH$_4$ flux due to the presence of living reed would be in contradiction to an experiment performed by Grünfeld and Brix (1999). They found a decrease in the CH$_4$ emissions of 34% with the addition of *Phragmites* to a submerged organic soil. Their explanation is that methanogenesis is reduced and CH$_4$ oxidation increased due to the transport of oxygen by *Phragmites* into the rhizosphere. In a soil without reed, the gas transport would be dominated by ebullition. Transport by ebullition is faster than internal plant transport, so that less of the produced methane is oxidized. Hendriks *et al.* (2010) found the opposite in a field study with water table differences and vascular wetland plants. A high water table and vascular plants showed higher methane emission than the same soil and water table without vascular plants. Kankaala *et al.* (2004) found a higher contribution of ebullition to the CH$_4$ flux in a less dense *Phragmites* shore zone (24 shoots m$^{-2}$) than in a dense area (78 shoots m$^{-2}$). The less dense *Phragmites*-zone showed threefold higher CH$_4$ emissions than the denser area. Koch *et al.* (2014) also found a negative correlation of methane fluxes with *Phragmites* abundance. Given this negative correlation, the high density of 68 shoots m$^{-2}$ at our site would suggest that total CH$_4$ flux would be lower compared to wetlands with lower densities. Our observed yearly CH$_4$ flux of 30 g m$^{-2}$ a$^{-1}$ is in the same range as Kankaala *et al.* (2004) found (20 to 50 g m$^{-2}$ a$^{-1}$) with similar dense reed vegetation, and indeed almost 3 times lower than the flux measured by Kim *et al.* (1998b) in a reed density of only 35 shoots m$^{-2}$. So even though our site has a relative low net CH$_4$ flux, it is likely that plant mediated gas transport during the growing season could lead to higher CH$_4$ emissions compared to the winter season if in both seasons the ebullition is reduced due to the plant density.

### 4.2 Effect of other environmental factors on CH$_4$

Another influence of the plants on the CH$_4$ fluxes could be the release of root exudates, which is closely linked to photosynthesis. Root exudates lead to an increase in substrate availability in the form of easily decomposable organic compounds, which can be used by methanogens to produce CH$_4$ (Aulakh, *et al.* 2001; Christensen *et al.*, 2003). There are studies that found positive correlations between radiation or net ecosystem production and CH$_4$ flux (Whiting & Chanton, 1993; Joabsson & Christensen, 2001), although there are also studies that found the opposite (Mikkelä *et al.*, 1995; Ström *et*

*al*., 2005). Ström *et al*. explain the negative correlation by the $CH_4$ oxidation rate, depending on the oxygen transport capacity of the plants. In general, we expect that there should be an effect of photosynthesis on the $CH_4$ flux, because plant photosynthates are an important carbon source for methanogens (Philippot *et al*., 2009). It is nevertheless hard to say how much the diurnal pattern is influenced by this. In our data we see that on average the maximum $CH_4$ flux appears almost at same time as the maximum $CO_2$ uptake. It is the question, however, if the response time between photosynthesis and excretion of root exudates to the production of $CH_4$ could be that fast. Most studies that relate root exudation to $CH_4$ flux/production did not measure in hourly resolution. Nevertheless, Aulakh *et al*. (2001) found a $CH_4$ production peak 1 day after adding root exudates to pre-incubated clay soils, whereas Ström *et al*. (2005) found emission peaks for $CH_4$ and $CO_2$ more than 2 days after adding labeled acetate to soil with wetland plants in monoliths. Another reasoning why we assume that the diurnal pattern is mainly caused by the internal pressurized convective flow of the plants, builds on the observation that there is still a diurnal pattern for $CH_4$ flux visible in October: photosynthesis has come to an end, but the plants are still (partly) alive.

During winter, the daily pattern in the $CH_4$ fluxes was no longer visible. Dead culms of reed are able to transport $O_2$ into the soil and $CH_4$ and $CO_2$ from the soil to the atmosphere, but only by diffusion (Brix, 1989). During the winter months, correlations of gas fluxes with environmental variables were low. Nevertheless, the highest correlation was with soil temperature. This suggests that soil temperature played the dominant role during this period.

Soil temperature influences microbial activity (Moore, 1994; Le Mer and Roger, 2001). It also influences respiration, which influences the availability of substrate needed for methanogenesis ($CO_2$, acetate) (Christensen *et al*., 2003). Therefore, an increase in temperature leads to higher emissions.

Water table height is known to have a large impact on $CH_4$ fluxes (Moore and Knowles, 1989; Aerts and Ludwig, 1997; Grünfeld and Brix, 1999; Updegraff *et al*., 2001), but only for non-flooded peatlands. In our case, the impact was small because the water table was always above surface level (5-40 cm) so that the soil remained anoxic.

**4.3 $CO_2$ flux patterns**

Also $CO_2$ fluxes exhibited clear diurnal and seasonal patterns. The fluxes were mainly influenced by the presence of green plants (high negative correlation with global radiation) and temperature changes. A similar diurnal and seasonal variation was observed in a *Phragmites*-wetland in North-East China based on eddy covariance measurements (Zhou *et al*., 2009). They also observed the highest $CO_2$ uptake in July with -13.6 g $CO_2$ m$^{-2}$ day$^{-1}$, which is lower than our measured uptake of -17.5 g $CO_2$ m$^{-2}$ day$^{-1}$, and a small release of $CO_2$ in winter, which is in the same range (2.6 g $CO_2$ m$^{-2}$ day$^{-1}$ vs. 2.7 kg $CO_2$ m$^{-2}$ day$^{-1}$) as our observations. The difference in July can be explained by the much higher soil and ambient temperature in the study of Zhou *et al*. (2009), which resulted in higher $R_{eco}$ relative to the increase in assimilation.

The $Q_{10}$ values based on soil temperature in the study of Zhou *et al*. (2009) are within the same range (4.1 in May to 1.8 in September/October) as what we found (3.93 in May-June to 1.53 September-October). This number is still higher than the

$Q_{10}$ value found by Mahecha *et al*. (2010) based on a global collection of FLUXNET data. A possible reason might be that we used soil temperature instead of air temperature. Soil temperature gave a much better fit in the regression and soil respiration is also expected to have a high contribution due to the high carbon content in the soil. But plant respiration contributes also a certain fraction, and is more dependent on air temperature. Due to the almost permanent water logging at the study site, the difference between the ranges of air night temperatures and soil night temperatures was particularly large.

**4.4 Ecosystem as carbon and GHG sink**

    The yearly $CO_2$ uptake was 894 g m$^{-2}$ a$^{-1}$ and the $CH_4$ emission 30 g m$^{-2}$ a$^{-1}$. The $CO_2$ uptake is almost 4 times higher than in a *Phragmites* wetland in China (Zhou *et al*., 2009). The difference could be explained by the lower temperature at our site, so that the respiration rate is lower. Our $CH_4$ flux is in the same range as at sites with similar *Phragmites* densities (see above). More in general, northern fens show a wide variation in $CH_4$ fluxes, from close to zero to 300 g $CH_4$ m$^{-2}$ a$^{-1}$,

depending on temperature, water table and vegetation cover, among others (Lai, 2009; Kayranli *et al*., 2010). Our site is at the lower range of that spectrum.

    Summing up $CO_2$ and $CH_4$ fluxes of our ecosystem leads to the net annual carbon uptake of 220 g C m$^{-2}$, which is 26% of the gross ecosystem production (GEP). It should be noted that the calculated respiration rate during daytime might be underestimated due to the plant-mediated gas transport. Brix *et al*. (1996) measured that around noon, 5 times more $CO_2$ was

350 transported from the soil to the atmosphere by *Phragmites* plants compared to the early evening. Because daytime respiration is only estimated from its nighttime relationship with soil temperature, respiration could be underestimated and therefore the GEP as well. This would mean that the percentage of the GEP stored in the system would be lower than is given above. It is however hard to say how much lower, since we cannot independently assess the respiration rate during daytime.

    The carbon uptake at our site is much higher than the 65 g C m$^{-2}$ (5% of the GEP) measured in the *Phragmites*-dominated

wetland in Northeast China (Zhou *et al*., 2009) ($CH_4$ is not considered). But it is only half as high as the uptake estimated for a *Phragmites*-dominated wetland in Denmark (550 g C m$^{-2}$ a$^{-1}$, 47% of GEP; Brix *et al*., 2001). The temperature during the growing season was lower than in the Chinese wetland, on average even 10 °C lower in July. Zhou *et al*. calculated a much higher $R_{eco}$, which may have caused the difference. The wetland described by Brix *et al*. (2001) has a similar $R_{eco}$ as ours, but a 30% higher GEP, which explains the diverging findings. Our measured uptake of 65 g C m$^{-2}$ a$^{-1}$ fits in the wide range of

measured carbon exchange in northern peatlands: from an uptake of 220 g C m$^{-2}$ a$^{-1}$ to a release of 310 g C m$^{-2}$ a$^{-1}$ (Strack, 2008). On the long term, the uptake of northern peatlands is on average between 20-50 g C m$^{-2}$ a$^{-1}$ (Strack, 2008). A longer measurement period may be needed however to ensure that C uptake at our site falls indeed into this range.

**5. Conclusions**

Our $CH_4$ fluxes show distinct diurnal cycles, but only in the period when living green plants were present. This strongly suggests that plant-mediated gas transport of (a convective transport in the stem of *Phragmites*) plays an important role regarding the emission of $CH_4$ from a natural fen site in the Federseeried, Southern Germany. The convective flow within the plant is probably not solely driven by the humidity gradient between the interior of the plant and ambient air (HIC theory). From our data it is more likely that global radiation plays a more significant role in creating a higher pressure inside the plant.

Our research site is in the measured year a sink for both carbon (-221 g C $m^{-2}$ $a^{-1}$) and greenhouse gases (-52 g $CO_2$-eq $m^{-2}$ $a^{-1}$). This is probably due to the high productivity of *Phragmites* plants, high water table and the relatively cold climate so that respiration rates are relatively low. Thereby, the low $CH_4$ emission compared to other *Phragmites* wetlands can be explained by the high plant density in our system, which could reduce ebullition.

In general, the role of wetland plants that can enhance gas transport, such as *Phragmites*, is important to consider for the determination of the impact of these wetlands on climate change. The role of environmental factors such as global radiation and relative humidity on the convective flow within *Phragmites* should be further investigated. This would be helpful to gain more knowledge about the contribution of the plant-mediated-transport to the net fluxes of $CH_4$ and $CO_2$.

**Acknowledgement**

We would like to thank Heidje Reinhard for her substantial contribution during the setup of the eddy covariance system and Giani Gangloff and Jamie Smidt for their technical assistance in the field. We also like to thank the Federal State of Baden-Württemberg (BWPLUS program) for their financial support that made this research possible.

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

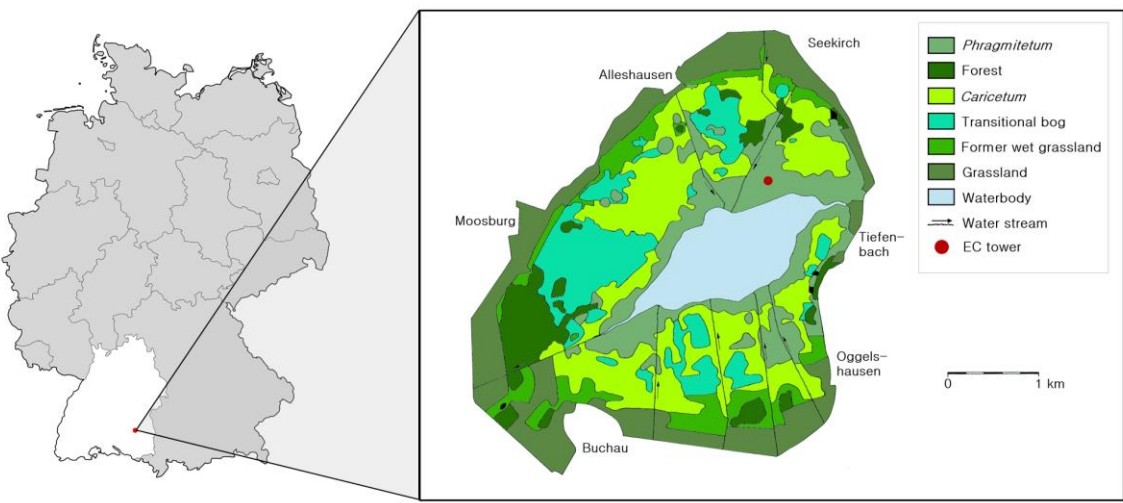

**Figure 1:** Land cover map of the most natural part of the nature conservation area Federseemoor (adapted from Grüttner and Warnke-Grüttner (1996)), located in the Federal State of Baden-Württemberg in Germany. The Eddy-Covariance (EC) tower was built northeast of the lake in the center of the largest reed area.

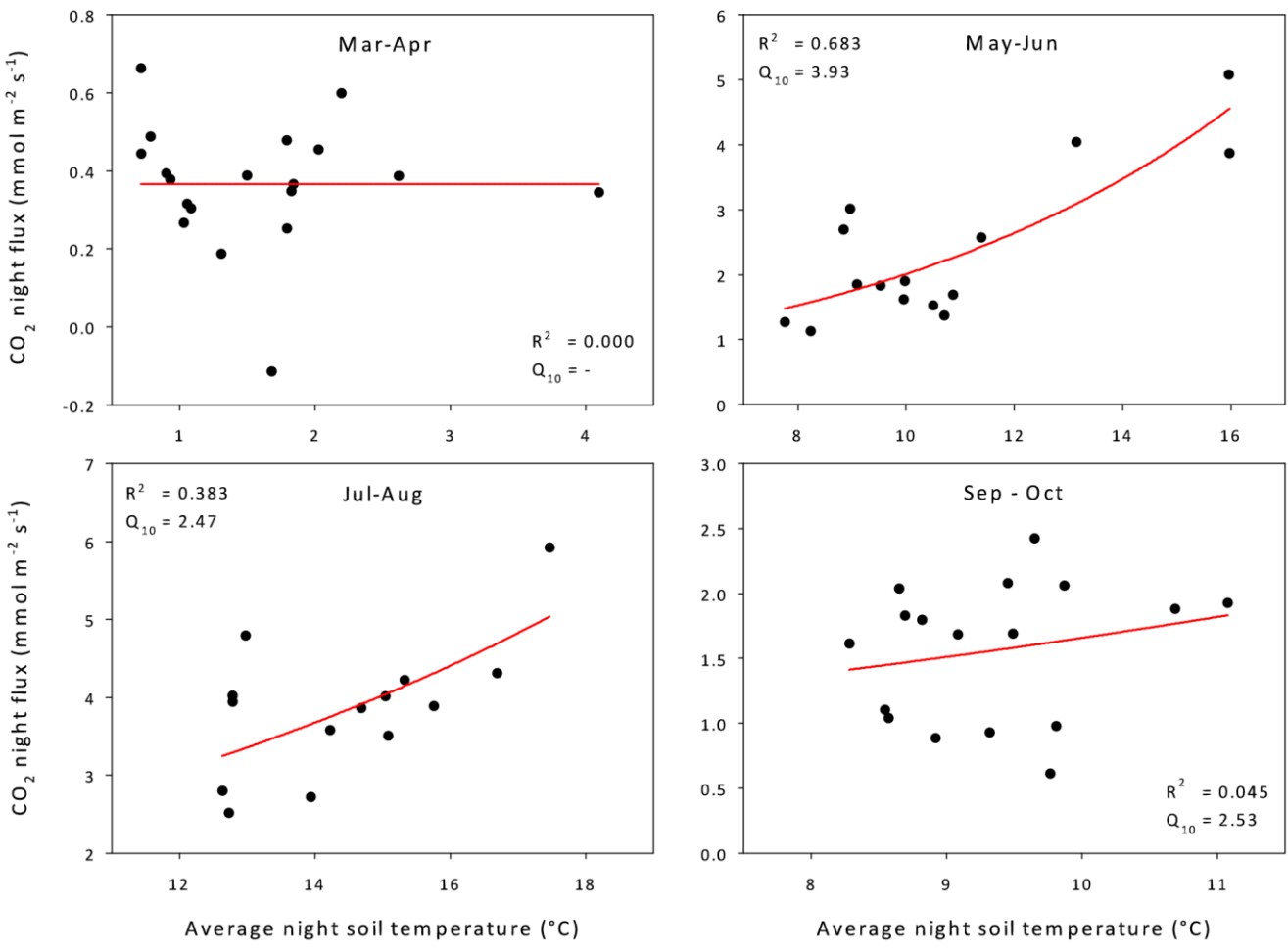

**Figure 2:** Average night fluxes for every 2 months during the growing season, against average soil night temperature (dots) with the regression model outcome (red line).

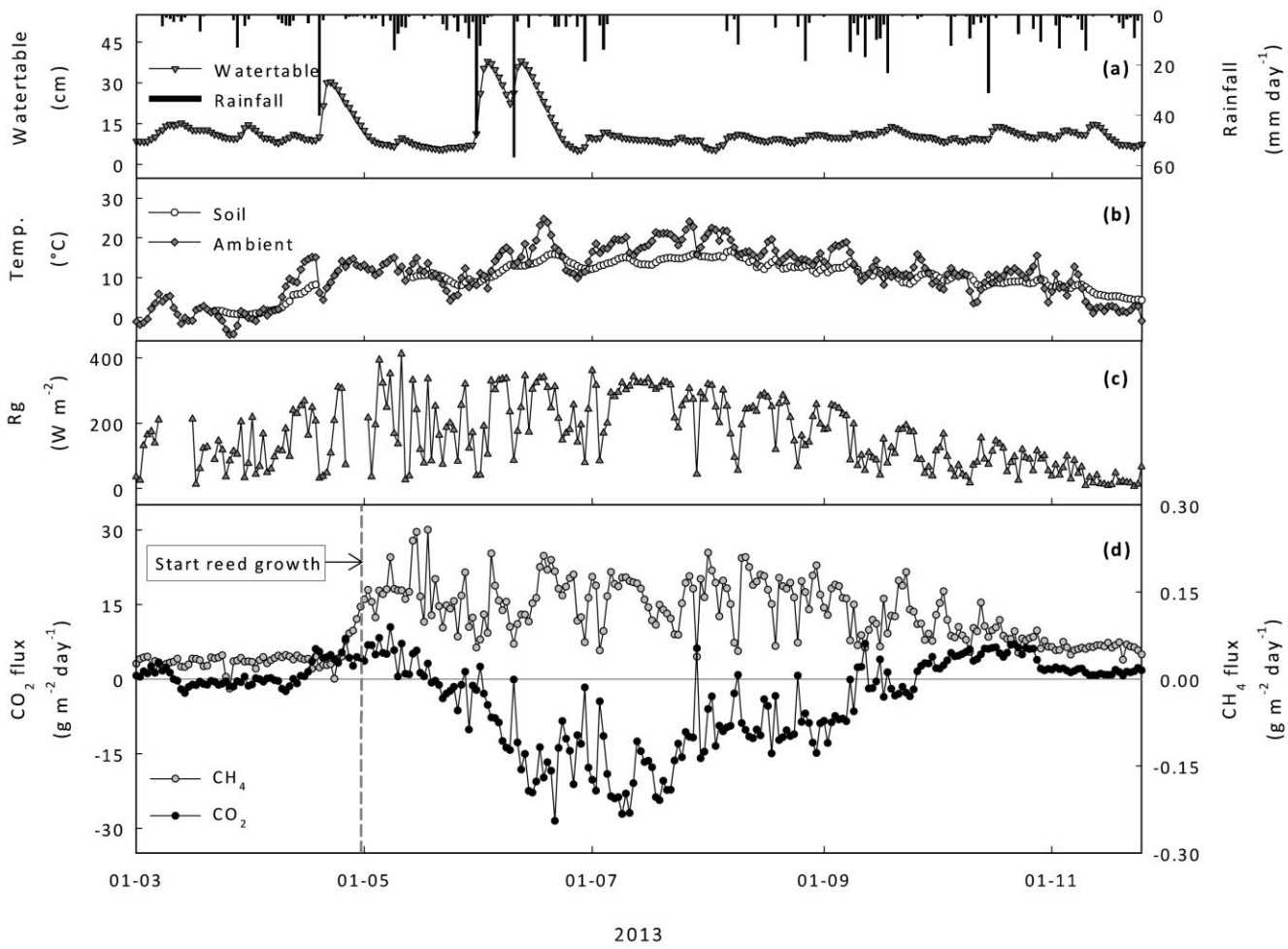

**Figure 3:** Daily averages of water table and rainfall (a), soil and ambient temperature (b), global radiation (Rg) (c) and $CO_2$ and $CH_4$ fluxes (d) in the period 01-03-2013 to 25-11-2013.

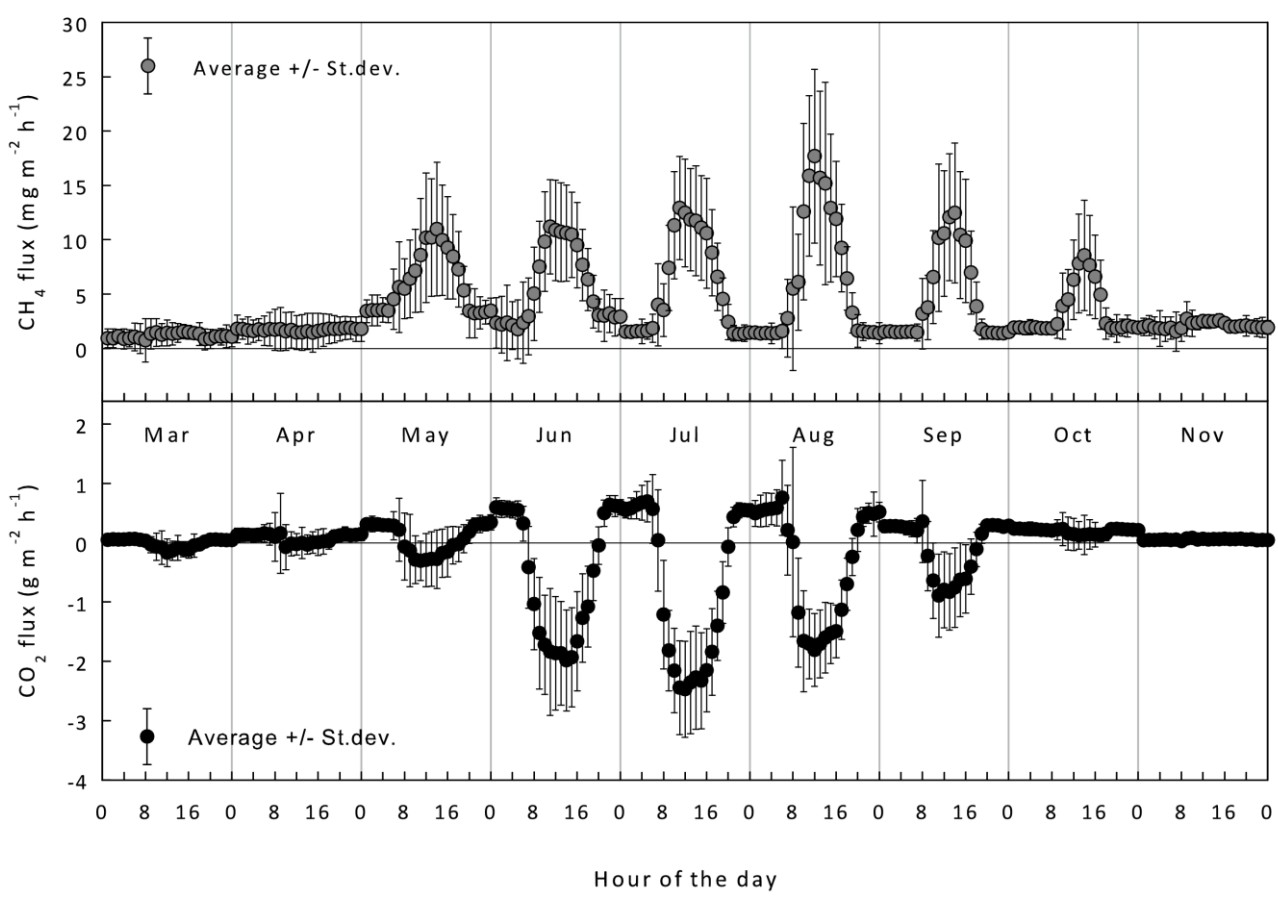

**Figure 4:** Diurnal cycles of CH$_4$ (above) and CO$_2$ (below) from March 2013 till November 2013. Each point represents the flux averaged over a specific hour of the day averaged over one month. Error bars denote standard deviations.

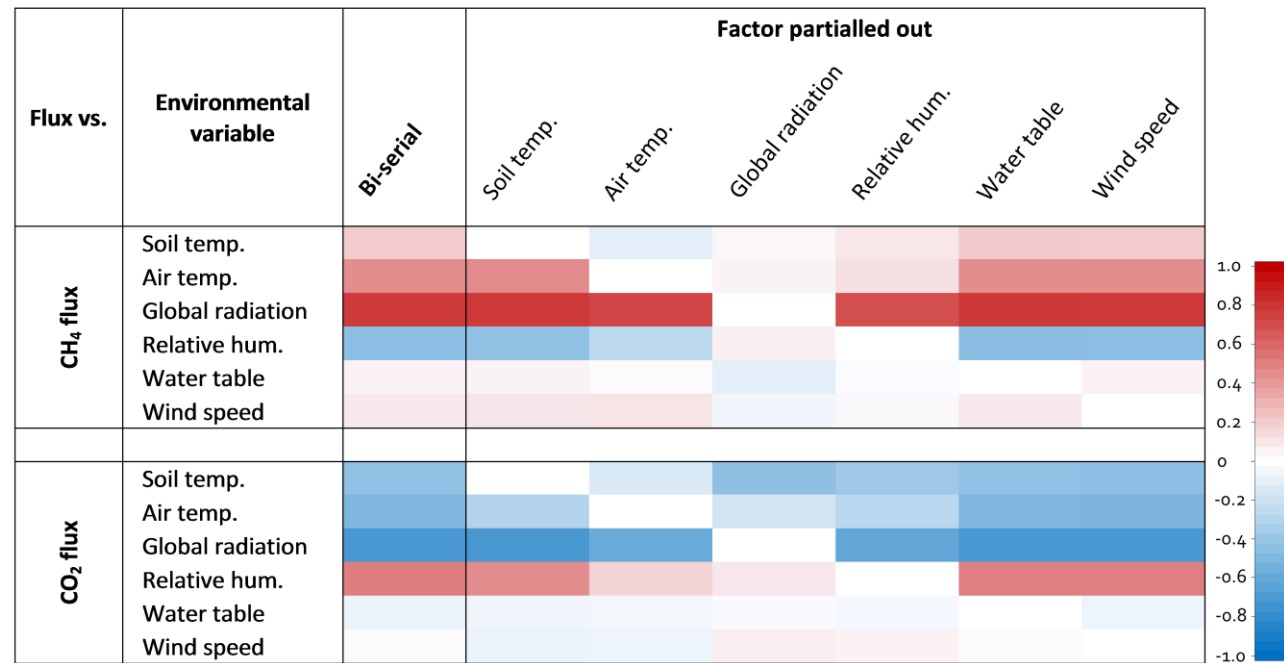

**Partial-correlation table**

**Figure 5:** Biserial and partial correlations between $CH_4$ flux (above) or $CO_2$ flux (below) and environmental variables within the growing season (May-October). Each variable is partialled out and the corresponding correlations with the other variables are shown in the same column. The darker the cells, the higher the correlation coefficient, with the red colors for positive correlations and blue for negative correlations.

**Table 1:** Model parameters and statistics of the $CO_2$ and $CH_4$ ARIMA (0,1,1) models.

| Model variable | CH$_4$ flux (g m$^{-2}$ day$^{-1}$) | | | CO$_2$ flux (g m$^{-2}$ day$^{-1}$) | | |
|---|---|---|---|---|---|---|
| | Coefficient ($\beta$, $\theta$) | t | P-value | Coefficient ($\beta$, $\theta$) | t | P-value |
| Rg (W m$^{-2}$) | $7.15.10^{-4}$ | 8.83 | <0.000 | -0.0859 | -5.42 | <0.000 |
| Rg$^2$ (W m$^{-2}$) | $-6.07.10^{-7}$ | -2.94 | 0.004 | $1.36.10^{-4}$ | 3.41 | 0.001 |
| Tsoil (°C) | | | | -0.950 | -2.56 | 0.011 |
| MA lag 1 | 0.388 | 5.41 | <0.000 | 0.671 | 11.2 | <0.000 |

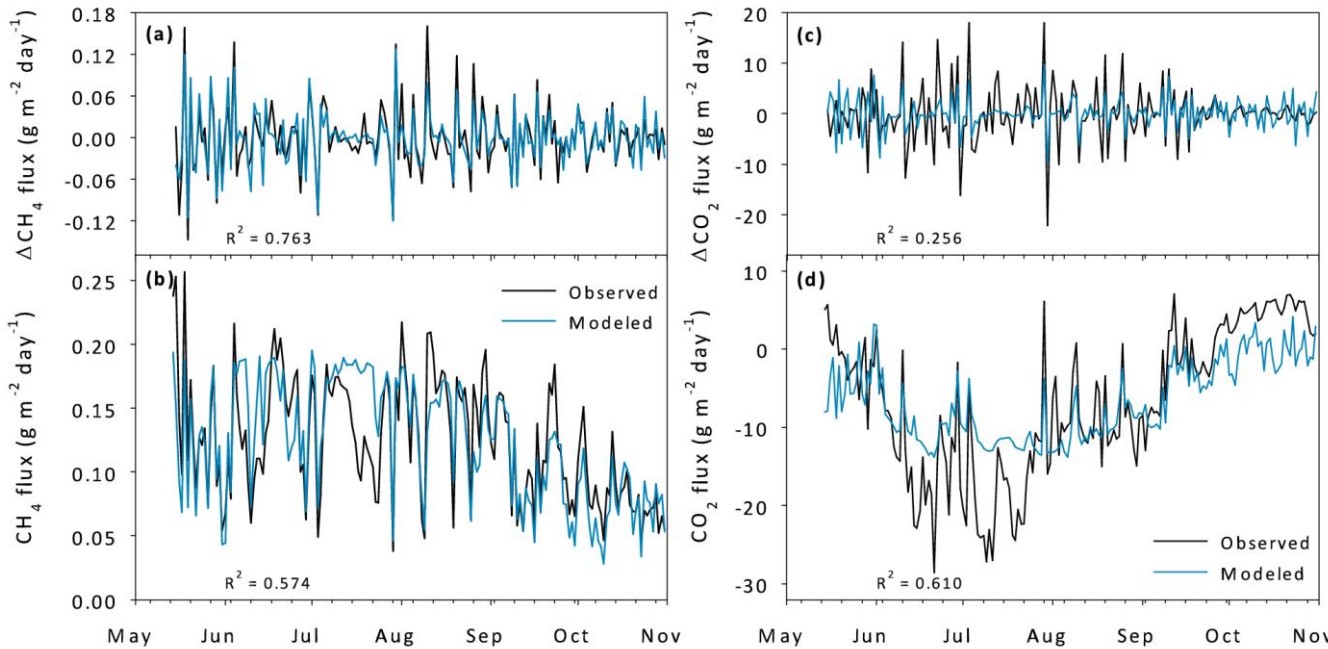

**Figure 6:** Observed and modeled daily fluxes of differenced data for $CH_4$ (a), original data for $CH_4$ (b), differenced data for $CO_2$ (c) and original data for $CO_2$ (d). The modeled data are created with an ARIMA(0,1,1) model, with Rg as explaining variable for $CH_4$ and Rg and Tsoil for $CO_2$. The error terms with the autoregressive part for the modeled data are not included in these graphs and in the coefficients of determination ($R^2$).

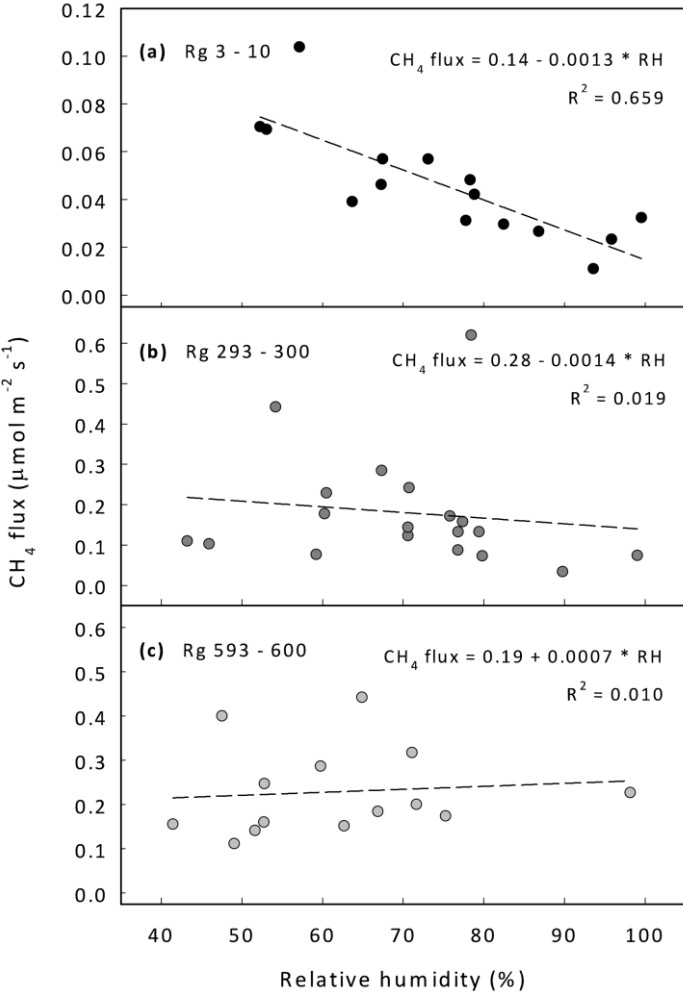

**Figure 7:** CH₄ flux plotted against relative humidity (RH). Data are selected by global radiation (Rg) in the range 3–10 W m⁻² (a), 293–300 W m⁻² (b) and 593–600 W m⁻² (c). Only data with corresponding air temperature between 10 and 20 °C are displayed.

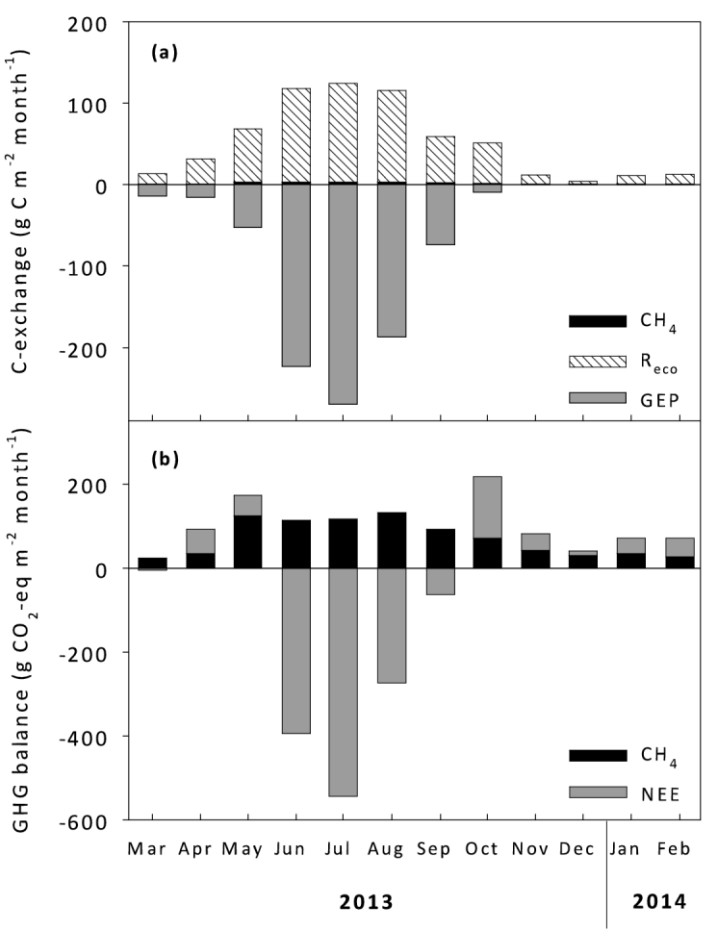

**Figure 8:** Monthly accumulated carbon fluxes from $CO_2$, divided in $R_{eco}$ and GEP, and $CH_4$ (a) and carbon fluxes in $CO_2$-equivalence from $CO_2$ (NEE) and $CH_4$ (b), where $CH_4$ fluxes are multiplied with the $GWP_{100}$ of factor 28 (IPCC 2013).

**Table 2:** Annual integrated flux, carbon balance and greenhouse gas (GHG) balance for $CO_2$, $CH_4$ and the sum of both for carbon and GHG balance.

| | **Integrated flux** (g m$^{-2}$) | **Carbon balance** (g C m$^{-2}$) | **GHG balance** (g CO$_2$-eq m$^{-2}$) |
|---|---|---|---|
| $CH_4$ | 30 | 23 | 842 |
| $CO_2$ | -894 | -244 | -894 |
| **Sum** | | **-221** | **-52** |