# Peer review of "The role of *Phragmites* on the CH4 and CO2 fluxes in a minerotrophic peatland in southwest Germany"

_Biogeosciences, 2016_

## Referee Comment (RC1) · T. Arkebauer (Referee) · 13 May 2016

The manuscript discusses research conducted in a wetland in southern Germany and is focused on quantifying CH4 and CO2 fluxes in relation to internal pressurization in Phragmites. Various environmental controls on pressurization in Phragmites are evaluated for their influence on the observed fluxes. In general, the manuscript is well-written and scientifically sound.

One of the concerns I have is due to the 2 month (or so) gap in the wintertime data. The authors discuss their gap filling methods (basically using a Fluxnet approach for filling in missing CO2 flux data) which seem like a reasonable approach due to the lack of published techniques for filling missing CH4 flux data. However, I believe their

presentation could be improved by providing error estimates for the reported fluxes, particularly the monthly and annual estimates presented in Figure 8 and Table 2. Although the missing data likely represent times of fairly small $CO_2$ and $CH_4$ fluxes, the length of the gap (2 months) is disconcerting when only a single year of flux data has been considered.

The discussion of the start of significant $CH_4$ emission and $CO_2$ uptake (lines 165 - 173) could be improved. The authors state that reed growth was initiated "by the end of April" and show a line on May 1st in Figure 3 to indicate this. The statements "From that moment, the reed plants assimilated $CO_2$ and daily $CO_2$ fluxes became negative. At the same time $CH_4$ fluxes rapidly increased." do not quite fit the data shown in Figure 3. It is apparent that the $CH_4$ flux increases some time before May 1st and the $CO_2$ flux does not go negative until some time in mid-May.

The presentation of monthly averages of diel fluxes (Figure 4), discussed in Section 3.2, may obscure finer details in the observation record. For example, from examination of Figure 4, the variability in the monthly averaged fluxes of both $CO_2$ and $CH_4$ increases at midday in for March (slightly) and April (more pronounced) whereas the authors state that "From May on, when new reed was present, a distinct diurnal pattern was established for both gases". Also, whilst not so easy to discern, it appears that the $CH_4$ flux peak may be slightly later in the day than the $CO_2$ peak but the authors state that "the highest negative fluxes for $CO_2$ and highest positive fluxes for $CH_4$ around noon".

The authors conclude that the lack of a pronounced effect of RH on the observed $CH_4$ fluxes (relative to the importance of global radiation) is noteworthy. This is a logical conclusion from the data presented in the manuscript. However, the range of ambient RH at the German site may have been smaller than the range of RH at the more semi-arid site in Nebraska reported by Kim et al. It is a bit difficult to say, though, since the present manuscript uses RH while Kim et al. reported vapor pressure deficits. Both studies measured these parameters above the canopy and, considering Armstrong and Armstrong's idea of the importance of the behavior of leaf sheath stomata, are

somewhat removed from the likely site of influence. Also, the influence of wind speed may not be as apparent at the German site compared to the Nebraska site but, again, without specifics it is hard to assess.

I especially liked the discussion on lines 349-355 for determining and interpreting ecosystem respiration in systems where plants pressurize.

Here are a few minor suggestions:

The use of 'diurnal' (i.e., daytime) should, in most cases, be changed to 'diel' (i.e., 24 hour). The Figure 6 caption has CO2 (no subscript for the 2). Line 270 has 'leave' instead of 'leaf'. Line 285 has 'photosynthetic' instead of 'photosynthetically'. Line 292 has 'stomata' instead of 'stomatal'.

---

## Referee Comment (RC2) · Anonymous Referee #2 · 9 Jul 2016

Van den Berg Biogeosciences Biogeosciences Discuss., doi:10.5194/bg-2016-122, 2016

The research aim was to clarify (1) how this plant-mediated gas transport influences the CH4 fluxes, (2) which other environmental variables influence the CO2 and CH4 fluxes, and (3) whether Phragmites peatlands are a net source or sink of greenhouse gases. CO2 and CH4 fluxes.

The authors used direct eddy covariance method to conduct their study. The method makes direct, but net flux measurements between the ecosystem and the atmosphere. To my mind it is the best technique for studying ecosystem greenhouse gas fluxes given the new generation of laser spectrometers and open path systems that run off

solar panels so scientists can study remote wetlands where there is no ac power. This is an emerging topic and field and ripe for studies like this with new data and new interpretations.

There are some noted strengths of this work including a year's budget of greenhouse gas fluxes from a peatland. These are important greenhouse gas hot spots and only in the past few years have there begun to be continuous records of fluxes from these systems. In the past most of the work was with chambers that were episodic in time and confined to small areas and sampled periodically. The eddy covariance method gives these investigators the ability to measure net and gross carbon fluxes and then relate gross carbon assimilation with methane fluxes. How cool can this be towards addressing attributions of biophysical mechanisms towards understanding methane production and transport.

With methane exchange it is important to know if the flux is due to bubble transport, diffusion through the water column or xylem transport. Here the authors attempt to study the route by which methane enters the atmosphere. And give us insight on the dominate mechanism for transport.

Introduction

The authors do a nice job reviewing the literature and capturing key papers like those by La Mer and Lai. There is also nice work by Moore (Moore, 1994), Megonigal (Megonigal et al., 2003) and by Brigham (Bridgham et al., 2006). But it may not be necessary to cite everyone.

Methods

The authors use the eddy covariance and open path licor 7700. This tunable diode laser system is state of art, works off line and has been well vetted, so I have confidence in fluxes exceeding 10 nmol m-2 s-1, as long as the correct density fluctuations corrections are made. I would like to hear more about flux detection limits, calibration and errors. They do a good job applying the standard tests for interpreting fluxes, following work by Foken and co workers.

As for gap filling while the Falge methods are standard for CO2 fluxes other methods should be applied for methane flux gap filling. Look to the work of Sigrid Dengel et al. and others for methane gap filling. I suggest the use of artificial neural networks to gap fill methane fluxes. This approach is gaining popularity by groups such as those led by Gil Bohrer and another by the author Cove Sturtevant, in JGR Biogeosciences.

The simultaneous measurements of carbon dioxide and methane and the partitioning of NEE into GPP is what I like about this paper. There is much strength and potential of doing such coupled research. The authors are using appropriate methods to partition NEE into GPP and Reco with qualifications.

The authors compute one Q10 value through the whole data set and come up with a non biological and indefensible Q10 value greater than 2 and near 4. While this may be ok for gap filling, it is wrong fundamentally and can be misused by modelers who may look for a Q10 from these data. From lab enzymatic studies we know Q10 value is near 2. We have learned from CO2 studies that the Q10 will be artificially high when and the basal rate of respiration changes with the season. So the basal rate must be adjusted with time; this is the main lesson from the Reichstein (Reichstein et al., 2005) paper and Mahecha (Mahecha et al., 2010) paper.

There may be difficulty in interpreting methane fluxes as the source distribution may be heterogenous. The authors need to supply us with information on the flux footprint climatology.

The authors use a biserial and muli correlation method to infer that stomata control the transport of methane because light is the strongest driver. While this is plausible and possible, they do not exclude the alternative hypothesis that photosynthesis primes methane production through root exudate to the rhizosphere. Remember we are looking for a balance between production and transport in interpreting fluxes.

The authors should look at both photosynthesis and transpiration as potential drivers of their methane fluxes, too. Many are showing that exudate from photosynthesis primes microbes that produce methane, so fluctuations in light could affect photosynthesis and methane production at certain time scales. This is an important alternative or complementary path and production mechanism. While I like much of the analysis and data I feel the authors need to do more than their simpler statistical analysis to nail down the answer. They have a rich and high quality dataset that merits deeper scrutiny. Doing this and it will be a first rate paper.

In sum I am worried about the attribution of causation of the plausible hypotheses. The authors inappropriate use linear regression models for a complex, nonlinear and multifactorial process is my biggest criticism. They are bound to misinterpret their data with such an antiquated statistical method. The Gil Bohrer team fitted their data with neural networks and looked at partial derivatives with environmental drivers to explain methane fluxes. More recently Sara Knox in a paper in JGR Biogeosciences used this method to study the controls of the environment on methane fluxes. The method seems to have much power. She and colleagues found superior description of their data using neural networks compared to a simple stepwise multi-linear regression model. At least the authors should do this. Remember we are trying to tease out the controls on fluxes that are modulated at different time scales by an array of different biophysical factors to different degrees. So the problem needs to be tackled with the best and most appropriate statistical methods. In addition, the field has advanced by introducing such methods as Granger Causality and Transfer Entropy to do a better job at linking methane and carbon fluxes with drivers such as light, temperature, humidity and photosynthesis (Hatala et al., 2012; Ruddell and Kumar, 2009). The authors have the dataset to apply these methods and I feel the work and interpretation would be stronger if they used them. These methods and tools are shared on the internet through MATLAB so they don't have to reinvent the wheel.

A nice side of this work is the computations of annual sums of carbon and methane

fluxes.

In sum I recommend publication after major revision. I think it has much potential to answer the important question they ask. The current paper is a good start. I only want them to aspire for better. Good luck.

References

Bridgham, S.D., Megonigal, J.P., Keller, J.K., Bliss, N.B. and Trettin, C., 2006. The carbon balance of North American wetlands. Wetlands, 26(4): 889-916. Hatala, J.A., Detto, M. and Baldocchi, D.D., 2012. Gross ecosystem photosynthesis causes a diurnal pattern in methane emission from rice. Geophysical Research Letters, 39(6): n/a-n/a. Mahecha, M.D. et al., 2010. Global convergence in the temperature sensitivity of respiration at ecosystem level. Science, 329(5993): 838-40. Megonigal, J.P., Hines, M.E. and Visscher, P.T., 2003. Anaerobic Metabolism: Linkages to Trace Gases and Aerobic Processes. In: H.D. Holland and K.K. Turekian (Editors), Treatise on Geochemistry. Pergamon, Oxford, pp. 317-424. Moore, T.R., 1994. Trace Gas Emissions from Canadian Peatlands and the Effect of Climatic-Change. Wetlands, 14(3): 223-228. Reichstein, M. et al., 2005. On the separation of net ecosystem exchange into assimilation and ecosystem respiration: review and improved algorithm. Global Change Biology, 11(9): 1424-1439. Ruddell, B.L. and Kumar, P., 2009. Ecohydrologic process networks: 1. Identification. Water Resources Research, 45(3): n/a-n/a.

---

## Author Comment (AC1) · 30 Jul 2016

We would like to thank referee T. Arkebauer for his positive notes about the paper as a whole, his interest in the discussion point about the respiration, and also for criticizing the preciseness of our data descriptions. Many thanks as well for all the language corrections, they are very helpful. Below we give our reply to all discussion points.

"One of the concerns I have is due to the 2 month (or so) gap in the wintertime data. The authors discuss their gap filling methods (basically using a Fluxnet approach for filling in missing CO2 flux data) which seem like a reasonable approach due to the lack of published techniques for filling missing CH4 flux data. However, I believe their presentation could be improved by providing error estimates for the reported fluxes,

[Figure]

particularly the monthly and annual estimates presented in Figure 8 and Table 2."

The 2-months data gap is indeed large and the error unknown. To assess the error we will create an artificial gap of the other 2 winter months and fill it using the same method. The comparison with the real data will provide an error guess, which will be included in the paper.

"The discussion of the start of significant CH4 emission and CO2 uptake (lines 165 - 173) could be improved. The authors state that reed growth was initiated "by the end of April" and show a line on May 1st in Figure 3 to indicate this. The statements "From that moment, the reed plants assimilated CO2 and daily CO2 fluxes became negative. At the same time CH4 fluxes rapidly increased." do not quite fit the data shown in Figure 3. It is apparent that the CH4 flux increases some time before May 1st and the CO2 flux does not go negative until some time in mid-May."

Reed growth starts at the 30th of April, so technically at the end of April. The line is just before the 1st of May. Yet, in the revised paper we will write the exact date to avoid confusion about this point. We will also be more precise about the statements like "CO2 fluxes became negative" by providing exact dates and more precise descriptions of the flux changes.

"The presentation of monthly averages of diel fluxes (Figure 4), discussed in Section 3.2, may obscure finer details in the observation record. For example, from examination of Figure 4, the variability in the monthly averaged fluxes of both CO2 and CH4 increases at midday in for March (slightly) and April (more pronounced) whereas the authors state that "From May on, when new reed was present, a distinct diurnal pattern was established for both gases". Also, whilst not so easy to discern, it appears that the CH4 flux peak may be slightly later in the day than the CO2 peak but the authors state that "the highest negative fluxes for CO2 and highest positive fluxes for CH4 around noon"."

There is some variation in the CH4 fluxes in March and April but it is not a clear diurnal

pattern. In March, the fluxes are a bit higher between 8:00h and 17:00h, but in April the lowest flux is at 14:00h and the highest at 21:00h. There is at least not a clear pattern compared to the pattern in the months that follow. For both fluxes, the peaks are always between 10:00h and 13:00h. $CH_4$ has only in May, September and October a later peak than $CO_2$. But in June and July, the peak is later for $CO_2$ (in August it is at the same time). So on average the statement we make holds. And the whole point of showing this graph is to get rid of the finer details, to be able to visualize this amazing strong diurnal pattern for $CH_4$ during the growing season. We will explain this point a bit better in the revised paper.

"The authors conclude that the lack of a pronounced effect of RH on the observed $CH_4$ fluxes (relative to the importance of global radiation) is noteworthy. This is a logical conclusion from the data presented in the manuscript. However, the range of ambient RH at the German site may have been smaller than the range of RH at the more semiarid site in Nebraska reported by Kim et al. It is a bit difficult to say, though, since the present manuscript uses RH while Kim et al. reported vapor pressure deficits. Both studies measured these parameters above the canopy and, considering Armstrong and Armstrong's idea of the importance of the behavior of leaf sheath stomata, are somewhat removed from the likely site of influence. Also, the influence of wind speed may not be as apparent at the German site compared to the Nebraska site but, again, without specifics it is hard to assess."

For the comparison with Kim et al. we will include the range of VPD. But in general, Armstrong and Armstrong stated that the humidity induced convective flow is negatively correlated with RH and will be close to zero when RH reaches 100%. In our data we only found this dependency with low radiation, especially when we take a look at Figure 7 where the radiation is more or less constant. We did not measure RH at vegetation level. Yet, neglecting temperature effects for the moment, RH will even be higher at ground level than above the canopy in such a wet, transpiring ecosystem. It is indeed hard to conclude something about the reasons why we did not find an

influence of wind speed in comparison to Nebraska. It is at least good that these measurements are done at different geographical locations, so that findings from one side are not generalized for all Phragmites systems over the world. We will add a few phrases to discuss this issue.

Please also note the supplement to this comment:
http://www.biogeosciences-discuss.net/bg-2016-122/bg-2016-122-AC1-supplement.pdf

―――――――――――――――――

---

## Author Comment (AC2) · 30 Jul 2016

We thank Referee #2 for his positive feedback on the Introduction and on the carbon and greenhouse gas budgets. His main criticism applies to the statistical method that we used to evaluate the relationship between fluxes and environmental variables. We intensively discussed his concerns in our group, and we consulted Prof. Hans-Peter Piepho who holds the chair of Biostatistics at our university. Please find below our reply to this and all other discussion points the referee gave.

"The authors use the eddy covariance and open path LICOR 7700. I would like to hear more about flux detection limits, calibration and errors."

We will extend our material and method with this additional information.

"As for gap filling while the Falge methods are standard for CO2 fluxes other methods should be applied for methane flux gap filling. Look to the work of Sigrid Dengel et al. and others for methane gap filling. I suggest the use of artificial neural networks to gap fill methane fluxes."

The gap filling method of Falge et al. (2001) is a well-established and widely used method in the eddy covariance community. We found that the correlations between environmental variables and the CH4 fluxes were at least equally strong as those with the CO2 fluxes. Therefore, we are convinced that for our CH4 fluxes the method of Falge et al. is as reliable as for CO2 fluxes, and we do not see the needfulness to change this well-established method to neural networks.

"The authors compute one Q10 value through the whole data set and come up with a non biological and indefensible Q10 value greater than 2 and near 4. While this may be ok for gap filling, it is wrong fundamentally and can be misused by modelers who may look for a Q10 from these data. From lab enzymatic studies we know Q10 value is near 2. We have learned from CO2 studies that the Q10 will be artificially high when and the basal rate of respiration changes with the season. So the basal rate must be adjusted with time; this is the main lesson from the Reichstein (Reichstein et al., 2005) paper and Mahecha (Mahecha et al., 2010) paper."

We are familiar with the paper of Reichstein et al. (2005). The reason we nevertheless decided to use the whole season for estimating the temperature dependency of Reco is because of the limited amount of available night data and the low variation in soil temperature due to the water saturated soil. This makes the regression less reliable. But indeed, in the field the dependency of CO2 on temperature is not only caused by enzymatic activities of microorganisms in the soil but also by, for instance, the available labile carbon which varies within the season. Although, the overestimation as described by Reichstein et al. will not be that high at our site (due to water-logged conditions, day time soil temperature is on average only 0.4 °C higher than night time soil temperature) we will nevertheless change our method and use Q10 values calculated for every 2 months (more or less the plant development stages and still supported by enough data points). The range of Q10 would then be from 1.53 (September-October) to 3.93 (May-June). This would on average still be a higher Q10 value than that found by Mehecha et al. (2010). A possible reason may be the use of soil temperature instead of air temperature. Soil temperature gave a much better fit in the regression and soil respiration is also expected to have a high contribution due to the high carbon content in the soil. But plant respiration will also contribute, and is more dependent on air temperature. The range of air night temperature is higher than that of soil night temperature. To make sure that the Q10 values are not falsely interpreted by modelers, we will consider renaming it and add a paragraph on the whole issue in the Discussion section.

"There may be difficulty in interpreting methane fluxes as the source distribution may be heterogenous. The authors need to supply us with information on the flux footprint climatology."

In general we can say that we have a very homogeneous footprint, with mainly Phragmites vegetation inside of it. Therefore, we do not expect a large variation of fluxes, for example, depending on wind direction. Yet, we agree that the information about the footprint was somewhat limited, and we will include more detailed information about it in the Material and Methods section.

"The authors should look at both photosynthesis and transpiration as potential drivers of their methane fluxes, too. Many are showing that exudate from photosynthesis primes microbes that produce methane, so fluctuations in light could affect photosynthesis and methane production at certain time scales. This is an important alternative or complementary path and production mechanism."

This is a very good and interesting point. We will add a paragraph on it in the Discussion section.

"The authors inappropriate use linear regression models for a complex, nonlinear and multifactorial process is my biggest criticism. They are bound to misinterpret their data with such an antiquated statistical method. The Gil Bohrer team fitted their data with neural networks and looked at partial derivatives with environmental drivers to explain methane fluxes. More recently Sara Knox in a paper in JGR Biogeosciences used this method to study the controls of the environment on methane fluxes. The method seems to have much power. She and colleagues found superior description of their data using neural networks compared to a simple stepwise multi-linear regression model. At least the authors should do this. In addition, the field has advanced by introducing such methods as Granger Causality and Transfer Entropy to do a better job at linking methane and carbon fluxes with drivers such as light, temperature, humidity and photosynthesis."

There are many different statistical methods available that could be used to analyze relationships between factors in datasets. Neural networks is an interesting tool for revealing connections between the input data (in our case the environmental variables) and the output data (the gas fluxes). Indeed, the advantage of neural networks is that it does not make detailed assumptions of the relation between input and output. It could be linear, but not necessarily. The main argument of the referee to use the neural network method is because of the non-linear correlations we may have in our data. We absolutely agree that a multiple linear regression might limit the outcome of the data analysis. There are also, however, disadvantages of neural networks compared to multiple regression models (Breiman 2001 and Warner & Misra 1996 give a good overview of the advantages and disadvantages of stochastic models/Neural Networks). Seen from a statistical point of view, neural networks have severe shortcomings. It is for instance not possible to test the significance of the coefficients (weights in the wording of neural networks). It is also not possible to compare different models and say which is statistically best. And the biggest disadvantage is that the functional relationships are hidden in a kind of black box. Knox et al. (2016) found a better fit of neural networks over linear models. But there are also examples of an almost equally good fit of a linear model compared to neural networks (Anagu et al. 2009). Methods like Granger Causality and Transfer Entropy are interesting for forecasting. But due to the auto-regression term included in the models, a large part of the variation that could be explained with environmental variables is removed. And we are not directly interested in forecasting, but in understanding the causalities in the system. Every method has advantages and disadvantages. We highly value the multiple regression-ARIMA method because of the fact that the detected relationships can be statistically analyzed and tested for significance. Therefore, we prefer to stick to the ARIMA method. We will discuss the pros and cons of our method and that of Knox et al. (2016) and related ones in the revised paper. We see as well, that the non-linearity in our data is a problem that could lead to false interpretation. To address these concerns, we will extend our ARIMA approach by testing nonlinear (quadratic and interaction) terms.

Added references: Anagu, I., Ingwersen, J., Utermann, J., Streck, T. (2009) Estimation of heavy metal sorption in German soils using artificial neural networks. Geoderma 152 104-112 Breiman, L. (2001) Statistical Modeling: The Two Cultures. Statistical Science 16(3) 199-231 Warner, B., Misra, M. (1996) Understanding Neural Networks as Statistical Tools. The American Statistician 50(4) 284-293

Please also note the supplement to this comment:
http://www.biogeosciences-discuss.net/bg-2016-122/bg-2016-122-AC2-supplement.pdf

---

## Author Response (AR1)

Changes in the manuscript were made according to our reply to the referees and are visible as red underlined text. Below we shortly describe these changes, followed by the changed manuscript.

**Referee 1**

5

We changed all the language suggestions, except for the change from diurnal to diel. Oxford dictionary gave the description for diurnal: Daily, of each day (diurnal rhythms). And that is how we use the word. We also found the term diurnal cycle in more scientific papers with the same meaning to the word.

We filled our 2 months data gap with the lookup table approach. The referee asked for an error estimation for this gap filled data. We suggested to fill an artificial gap of 2 months and compare this with the measured data. To make the comparison as reliable as possible, we wanted to make the artificial gap and the lookup table from data outside the growing season. Therefore it was not possible to make an artificial data gap of 2 months, but only of 1 month. An error estimation was made

10 Therefore it was not possible to make an artificial data gap of 2 months, but only of 1 month. An error estimation w with these data.

According to the referee, the description of the data in paragraph 3.2 was not precise enough. We added some information to avoid confusion about our data description.

The referee thought that the dependency of our  $CH_4$  flux on relative humidity was hard to compare with the side of Kim *et al.* (1998) because Kim et al. used VPD. We nevertheless did not change our data to VPD, or added information about it, because we mainly compare our data with Armstrong and Armstrong, who directly measured the dependency of Humidity Induced Convection on relative humidity. We added some phrases in the discussion to make this point clearer. We also did not compare the wind speed to that of Kim *et al.* since they found a correlation in night data, while we did not separate our data into day and night for the statistical analyses. We did compare the magnitude of our flux data with that of Kim *et al.*

**20 Referee 2**

The referee wanted to hear more technical information about the LI-7700. We extended the material and method with this information.

25

The  $Q_{10}$  value in our manuscript was quite high. The referee argued that this is due to the calculation of this  $Q_{10}$  value over the whole season. We therefore calculated  $Q_{10}$  values for periods of 2 months during the growing season and used these values to recalculate the ecosystem respiration. We changed this part in the material and methods, including figure 2, and changed figure 8 with the updated data. We discussed the newly calculated  $Q_{10}$  values (paragraph 4.3) and also considering the work of Mahecha *et al.* (2010), where the referee is referring to. The referee requested more information about the flux footprint. We added this information in paragraph 2.2 and 2.3.

The referee suggested that we should consider the connection of photosynthesis, root exudates and methane production. This

30 means that CH4 fluxes could be light dependent due to this connection. We did some more literature research and discussed the outcome in paragraph 4.2.

The main criticism of the referee was our statistical methods. We used a linear model, even though it was not clear if the relationships are linear. Therefore we did the ARIMA analysis again, with the addition of quadrats to create a polynomial regression. With a polynomial function it is also possible to reveal different kinds of non-linear relationships. It appeared that

35 the models slightly improved with a second order of polynomial relation with radiation for both fluxes. Therefore, the models were changed (see paragraph 2.6) and the new results are shown in paragraph 3.3, including table 1 and figure 6.

[revised manuscript text omitted]

**3.2 Diurnal pattern**

245

gases are presented in hourly resolution (Fig. 4). CO2 shows a weak diurnal pattern in the months of March and April, while 220 there is no clear pattern visible for  $CH_4$ . From May on, when new reed was present, a distinct diurnal pattern was established for both gases, with the highest negative fluxes for  $CO_2$  and highest positive fluxes for  $CH_4$  around noon between 10:00h and 13:00h. Over the whole growing season, the daily maximum for  $CH_4$  and  $CO_2$  flux was on average 15 and 30 minutes, respectively, earlier than the radiation maximum. The highest midday to night difference for  $CH_4$  was observed in August with, on average, a midday flux of 15.7 mg CH4 m-2 h-1 and a night flux of 1.41 mg CH4 m-2 h-1. These values differ by a factor of 11. The highest uptake of CO2 around noon occurred in July (2.36 g CO2 m-2 h-1). Also in this month the highest 225 night flux was observed, on average a release of 0.629 g  $CO_2$  m-2 h-1.

To see how diurnal cycles of both  $CO_2$  and  $CH_4$  fluxes change over the season, the monthly averaged diurnal fluxes of both

The diurnal pattern of CO2 disappeared in October. From November on, only positive fluxes were measured. The diurnal pattern of CH4 continued one month longer and almost vanished in November.

**3.3 Factors affecting the fluxes during growing season**

230 Fig. 5 shows the results of the partial correlation analysis of the half hourly data for the growing period (May-October).  $CH_4$ flux shows the highest correlation with global radiation, followed by relative humidity and air temperature. The biserial correlation between  $CH_4$  fluxes and global radiation changes very little, no matter which other factor is partialled out. This suggests that global radiation is the most important factor influencing  $CH_4$  fluxes. The high biserial correlation of  $CH_4$  flux with relative humidity and air temperature, decreases considerably when global radiation is partialled out. This means that 235 the correlations of relative humidity and air temperature with the  $CH_4$  flux are based to a large extent on their correlation with global radiation. It is notable that the correlation of the  $CH_4$  flux with soil temperature is small. The correlation even becomes negative when air temperature is partialled out. During the winter period, results differ:  $CH_4$  flux correlates most with soil temperature (r=0.371), followed by water table height (r=0.222) (data not shown).

The correlation table for  $CO_2$  fluxes shows the same pattern, but inverse of  $CH_4$ , except that correlations with air and soil 240 temperature are higher than those of CH4.

The impact of environmental factors on the daily fluxes of  $CH_4$  and  $CO_2$  fluxes was evaluated by regression analysis in the framework of the ARIMA approach. An ARIMA(0,1,1) model was found to be suited to model the flux time series of both  $CH_4$  and  $CO_2$  (Table 1). Global radiation turned out to be the only regressor with a statistically significant impact (P < 0.05) on the CH4 fluxes, and global radiation and soil temperature on the CO2 fluxes. A second order polynomial model describes the relation between global radiation and CH4 flux as well as CO2 flux best. In the CO2 model, the addition of soil

temperature as a linear regression term was giving the best model results. Other environmental factors, such as relative humidity and air temperature, also covary with the fluxes, but their possible impact on the fluxes cannot be determined, because it is screened due to their correlation with global radiation and soil temperature. After differencing the data, the resulting models for  $CH_4$  and  $CO_2$ , are given by Equations (2) and (3)

$$\quad \Delta CH_4 f lux_t = \beta_1 * \Delta Rg_t + \beta_2 * \Delta Rg_t^2 + e_t - \theta e_{t-1}$$
(2)

$$\Delta CO_2 flux_t = \beta_1 * \Delta Tsoil_t + \beta_2 * \Delta Rg_t + \beta_3 * \Delta Rg_t^2 + e_t - \theta * e_{t-1}$$
(3)

where  $\Delta$  is the differencing operator ((e.g.,  $\Delta CH_{4,t} = CH_{4,t} - CH_{4,t-1}$ ),  $\beta$  a regression coefficient,  $\theta$  the weight of the moving average (MA) term and  $e_t$  the residual error term that is assumed to be independently normally distributed (white noise). The coefficient of determination (R2) of the CH4 model with differenced data is 0.7770.79. Without the error term ( $e_t - \theta_{t-1}$ ), R2 is 0.750.76. In case of CO2, the model performance is much lower (R2 = 0.430.47 with error term and R2 = 0.200.26 
[revised manuscript text omitted]